# Emotional Response Inhibition: A Shared Neurocognitive Deficit in Eating Disorder Symptoms and Nonsuicidal Self-Injury

**DOI:** 10.3390/brainsci10020104

**Published:** 2020-02-15

**Authors:** Kenneth J. D. Allen, M. McLean Sammon, Kathryn R. Fox, Jeremy G. Stewart

**Affiliations:** 1Department of Psychology, Oberlin College, Oberlin, OH 44074-1024, USA; msammon@oberlin.edu; 2Department of Psychology, University of Denver, Denver, CO 80210-4638, USA; kathryn.fox@du.edu; 3Department of Psychology, Queen’s University, Kingston, ON K7L 3N6, Canada

**Keywords:** binge eating, body image, cognitive control, compulsive behavior, eating disorders, emotional regulation, impulsive behavior, non-suicidal self-injury, self-injurious behavior, urgency

## Abstract

Eating disorder (ED) symptoms often co-occur with non-suicidal self-injury (NSSI). This comorbidity is consistent with evidence that trait negative urgency increases risk for both of these phenomena. We previously found that impaired late-stage negative emotional response inhibition (i.e., negative emotional action termination or NEAT) might represent a neurocognitive mechanism for heightened negative urgency among people with NSSI history. The current study evaluated whether relations between negative urgency and ED symptoms similarly reflect deficits in this neurocognitive process. A total of 105 community adults completed an assessment of ED symptoms, negative urgency, and an emotional response inhibition task. Results indicated that, contrary to predictions, negative urgency and NEAT contributed independent variance to the prediction of ED symptoms, while controlling for demographic covariates and NSSI history. Worse NEAT was also uniquely associated with restrictive eating, after accounting for negative urgency. Our findings suggest that difficulty inhibiting ongoing motor responses triggered by negative emotional reactions (i.e., NEAT) may be a shared neurocognitive characteristic of ED symptoms and NSSI. However, negative urgency and NEAT dysfunction capture separate variance in the prediction of ED-related cognitions and behaviors, distinct from the pattern of results we previously observed in NSSI.

## 1. Introduction

Nonsuicidal self-injury (NSSI), deliberately self-inflicted bodily harm without suicidal intent, frequently co-occurs with disordered eating patterns [1]. A recent meta-analysis found that over 27% of patients diagnosed with eating disorders (EDs) endorsed a lifetime history of NSSI [2], and conversely, up to 55% of individuals who engage in NSSI report some disordered eating activities [3,4]. NSSI and dysregulated eating behaviors (e.g., binge eating, food restriction, and purging) have been described as “direct” and “indirect” forms of self-injury, respectively [5]. These proposed categories refer to the distinct temporal relationships between different self-injurious behaviors and resulting physical harm as well as the extent to which that harm is intentional; NSSI was originally conceptualized as unique from ED behaviors in that physical harm is both immediate and deliberate. However, recent work suggests that both forms of self-injury can involve a desire to cause physical damage to oneself in the moment and over time, in addition to some degree of suicidal ideation [1]. 

Some researchers have proposed that emotion dysregulation and impulsivity may contribute to the observed comorbidity in these clinical phenomena [6,7,8,9,10,11]. Emotion dysregulation is a core feature of diverse self-injurious behaviors [12,13,14,15,16] and a putative non-specific marker of general vulnerability for psychopathology [17,18]. Consistent with this notion, the most commonly reported function of NSSI and ED behaviors is to reduce negative affect [19,20,21,22,23,24,25]. Ecological momentary assessment corroborates the idea that direct and indirect self-injury are both motivated by a desire to alleviate unpleasant emotional states, as increased negative affect proximally predicts episodes of NSSI, dysregulated eating, and compensatory behaviors [26,27,28,29,30,31,32].

Much like emotion dysregulation, impulsivity is implicated in various psychiatric disorders and self-injurious behaviors. This multifaceted construct encompasses several subfactors, including impulsive personality traits and impulsive behavior, or motor impulsivity. Impulsive behavior can be further divided into impulsive action and impulsive choice, which reflect inhibitory control and decision-making deficits, respectively. Impaired inhibitory control and consequently, impulsive action, are aspects of altered neurocognition broadly involved in suicidal thoughts and behaviors [12]. Impulsive personality traits, especially negative urgency, are also associated with NSSI [11,33,34,35,36] and dysregulated eating [36,37,38,39,40]. Negative urgency is a transdiagnostic personality characteristic that refers to individual differences in the tendency to act impulsively in response to negative affect [41]. Heightened negative urgency among people who engage in self-injurious behaviors suggests that these behaviors (a) serve to regulate aversive affect and (b) reflect impulsive acts precipitated by distress. Negative urgency is also associated with self-reported emotion regulation difficulties in NSSI and ED symptoms [42,43] and might thus represent an area of overlap between facets of emotion dysregulation and impulsivity common to both NSSI and disordered eating [10,11,12,33,34,35,36,42,43,44]. Accordingly, NSSI and disordered eating may frequently co-occur due to shared tendencies to react impulsively to distress, given the desire to rapidly reduce negative affect that promotes maladaptive coping strategies to “escape” these undesirable feelings, e.g., [10,11,12,25,26,27,28,29,30,31,32,33,34,35,36,42,43,44,45].

Elevated negative urgency may be a consequence of inhibitory control deficits that specifically arise in negative emotional contexts. For example, NSSI is associated with risky decision-making in response to critical feedback (eliciting negative affect) such that individuals with NSSI history are more likely to make impulsive choices during negative mood, but not necessarily in its absence [46]. Inhibitory control comprises three stages [47]: (1) Interference inhibition; (2) action restraint or suppression (early response inhibition); and (3) action cancellation or termination (late response inhibition). At the neurocognitive level, negative urgency is most closely linked to impaired response inhibition, implicating the second and third stages of inhibitory control in the expression of this trait. Despite this link, performance on the majority of behavioral and neuropsychological tasks measuring impulsivity are poorly or only modestly associated with negative urgency and other self-reported impulsive traits [48,49,50,51]. Accumulating evidence nuances our understanding of this relationship, suggesting that negative urgency may be more strongly tied to *emotional* response inhibition, particularly the ability to inhibit motor impulses driven by negative affect [52,53,54,55,56]. Allen and Hooley [53] recently found that deficit in late negative emotional response inhibition (Stage 3 of inhibitory control over emotional impulses, or “affective control”; see [12]) accounts for common variance in the relationship between negative urgency and NSSI history. We refer to this specific neurocognitive process as negative emotional action termination (NEAT): the ability to “cancel” or “terminate” an ongoing motor response triggered by negative affective reactions [53]. Participants with and without NSSI histories completed an emotional stop-signal task in that prior study [53] to replicate findings from earlier work [52]. Our findings indicated that NSSI is associated with difficulty inhibiting behavioral responses motivated by negative emotional reactions (once those responses are initiated), i.e., worse NEAT [52,53]. This effect is specific to the termination of ongoing motor impulses, in contrast to earlier stages of affective control requiring response withholding (i.e., emotional action suppression) [53] or inhibition of distracting emotional interference [57,58]. Subsequent work further suggests that late emotional response inhibition impairment might increase NSSI risk during real-world episodes of heightened negative affect and urgency [59].

We accordingly conceptualize NEAT impairment as a neurocognitive mechanism for heightened negative urgency in NSSI [53]. Studies have yet to address the underpinnings of dispositional negative urgency in disordered eating, despite substantial evidence indicating a comparable role for this trait in the development of EDs [34,35,36,37,38,39,40,43,44]. The present investigation sought to address this gap. Our primary aim is to examine whether impaired NEAT is uniquely associated with ED symptoms independent of negative urgency, or whether these factors account for common variance in a similar pattern to what we previously observed for NSSI [53]. We hypothesized that after controlling for NSSI history, poor NEAT would account for overlapping variance in negative urgency and ED symptoms in a community sample of adults. As a secondary hypothesis, we further predicted that NEAT would be more strongly associated with dysregulated eating behaviors (i.e., restrictive food intake, binge eating, and compensatory acts) relative to cognitive symptoms of EDs (i.e., eating-, weight-, and shape-related concerns). Although most participants included in these secondary analyses (*n* = 88; see below) were originally recruited on the basis of (presence/absence of) lifetime NSSI engagement, the substantial variation in ED symptoms among those with and without NSSI histories allow us to draw preliminary conclusions regarding whether NEAT serves as a shared neurocognitive mechanism for elevated negative urgency across these distinct forms of “direct” and “indirect” self-injurious behaviors.

## 2. Materials and Methods

Adult participants (18+) were recruited via (1) online postings and (2) printed advertisements in the community surrounding a large research university in the northeastern United States. Potential participants completed a web screening that collected demographic, psychiatric, and eligibility information. Inclusion criteria included English proficiency, no concussion history, and no impairments in motor ability, hearing, or vision. Participants provided informed consent prior to participation and received monetary or course credit compensation following completion of the study protocol. All study procedures were approved by the local university Institutional Review Board (IRB16-1592).

A subset of participants in this study were recruited for a lifetime history of NSSI; results comparing these individuals to participants who met inclusion criteria for a healthy control group (also included in the following analyses) are reported in previously published work; see [52]. Participants in that study were divided into two groups: those who report a history of at least one lifetime NSSI episode using any method (confirmed via semi-structured clinical interview, *n* = 45) and healthy control adults who reported no history of NSSI, suicidality, psychiatric treatment, or psychological problems (*n* = 43). Thus, the present study analyzed data from 17 individuals who did not endorse a lifetime history of NSSI nor met criteria to qualify for inclusion in a healthy control group, as well as 88 participants whose data were additionally reported in Allen and Hooley [45]. The total sample (*n* = 105) recruited for the larger study reported here comprises 45 participants who endorsed lifetime NSSI engagement and 60 participants who did not.

We used the Self-Injurious Thoughts and Behaviors Interview (SITBI) [60] to evaluate NSSI history. This semi-structured interview consists of 169 items to assesses the presence, frequency, and other characteristics (e.g., methods) of suicidal and non-suicidal self-injurious thoughts and behaviors, including suicide ideation, plans, gestures, attempts, and NSSI. The SITBI has acceptable test-retest reliability over a six-month period (mean κ = 0.70, intraclass correlation coefficient = 0.71), interrater reliability (mean κ = 0.99, *r* = 1.0), and construct validity [60].

The Eating Disorder Examination-Questionnaire (EDE-Q) [61] was used to assess ED symptoms and associated behaviors. The EDE-Q is a 22-item self-report measure derived from a structured clinical interview, which includes a global scale and subscales that gauge cognitive symptoms of EDs (eating concerns, shape concerns, and weight concerns), in addition to ED behaviors, including a restraint subscale (c.f., restrictive food intake) and items that evaluate frequency of past-month overeating, loss-of-control (LOC) eating, binge eating, vomiting, laxative use, and compulsive exercise. A score at or above 2.3 on the EDE-Q global scale indicates the likely presence of a diagnosable ED with considerable sensitivity and specificity [62].

Participants completed the UPPS-P [63] to assess different cognitive and behavioral facets of impulsivity. The UPPS-P is a 59-item self-report measure that includes five scales that evaluate the following impulsive personality traits: negative urgency, positive urgency, (lack of) perseverance, (lack of) premeditation, and sensation-seeking. The present analyses focus on the negative urgency scale, which measures the dispositional tendency to act impulsively in response to negative affective states; positive urgency indexes the corresponding tendency to act impulsively when experiencing *positive* effect. Perseverance captures individual differences in the tolerance for boring or difficult tasks; premeditation assesses the proneness to deliberate before taking action; and sensation-seeking is a metric for the propensity to pursue exciting, fun, and/or novel experiences. Participants rate each UPPS-P item on a 4-point Likert-type scale according to how much they agree or disagree (*1* = *Agree strongly* to *4 = Disagree strongly*) with statements such as (for negative urgency): “*When I feel bad, I will often do things I later regret in order to make myself feel better now*” and “*I often make matters worse because I act without thinking when I am upset*”. The UPPS-P demonstrates convergent and divergent validity in addition to acceptable test-retest reliability [63,64,65,66], e.g., *r* = 0.86 over a follow-up period of ten days [66]. We also found high internal consistency of responses to negative urgency items in the present sample, Cronbach’s α = 0.87.

We evaluated the emotional response inhibition, specifically *negative emotional action termination* (NEAT), using an Emotional Stop-Signal Task (ESST) [52,53]. In this task, participants were asked to quickly and accurately decide whether an image is “pleasant” or “unpleasant” and respond with a corresponding keypress. Images for the ESST were acquired from the International Affective Picture System [67] and included 12 images from positive, neutral, and negative valence categories (Neutral: 2102, 2215, 2280, 2305, 2385, 2396, 2411, 2440, 2480, 2516, 2840, 8312; Positive: 1340, 2045, 2075, 2091, 2209,2550, 4614, 5470, 5831, 8190, 8200, 8470; Negative: 2053, 2205, 2456, 2800, 2900, 3350, 6370, 6821, 9040, 9417, 9800, 9810) with equivalent valence intensity and arousal ratings. The task includes four blocks with 192 total trials that include randomly presented images from all three valence categories. 

Following original Stop-Signal Task procedures [68], 25% of trials (*n* = 48) included an auditory tone or “stop-signal.” Participants were asked to inhibit emotional reactions and accompanying motor (keypress) responses during trials in which a stop-signal is presented. These stop-signals occur with a variable delay after image presentation (50–1150 millisecond) that is adjusted in a staircase fashion (with 50 millisecond increments) according to participant performance, i.e., failed inhibition during a stop trial results in a shortened delay on the subsequent stop trial. This adjustment is meant to produce total commission error (i.e., “false alarm”) rates that approximate 50%. The primary dependent variable in the following analyses was NEAT, a metric of late-stage *negative* emotional response inhibition, i.e., difficulties terminating initiated behavioral responses driven by negative affect. We operationalize NEAT as the percentage of false alarms reflecting failures inhibiting negative emotional reactions, i.e., the proportion of commission errors with negatively valenced responses relative to total commission errors; see [53]. We therefore controlled for ESST accuracy (i.e., the percentage of images “correctly” classified as positive or negative) in regression analyses to confirm that effects of NEAT were not solely due to a negative response bias, a general tendency to react negatively to images in the absence of inhibitory demands.

We first calculated a composite variable (EDE-Q compensatory behaviors) as a summary score of the items assessing frequency of past-month vomiting, laxative use, and compulsive exercise, given the low endorsement of each specific behavior. After examining demographic variables to identify potential covariates for our primary regression analyses, we conducted a series of bivariate non-parametric correlations to examine associations among negative urgency, NEAT, NSSI history, EDE-Q scales, and specific EDE-Q items, i.e., frequency of past-month overeating, loss-of-control (LOC) eating, binge eating, and compensatory behaviors. We then performed a set of hierarchical multiple regression analyses examining the incremental contribution of NEAT beyond negative urgency in predicting EDE-Q outcome variables after controlling for NSSI history, ESST accuracy, and relevant covariates. We used linear regression models to evaluate EDE-Q scales (global, eating concerns, shape concerns, weight concerns, and restraint) and zero-inflated negative binomial regression models for the EDE-Q behavioral items, based on current recommendations in the field; see [69]. Section 3.4. includes results from count model portions of these non-linear regressions, which were of primary interest; please see Appendix A for zero-inflated logistic model results (Appendix A). Two participants were excluded from correlations and regressions based on ESST performance (outliers based on excessive omission errors) and a third participant had incomplete UPPS-P data; accordingly, *n* = 102 participants were included in those analyses.

## 3. Results

### 3.1. Demographics and ED Symptoms

The majority of participants identified as female, heterosexual, white, and single (never married; see Table 1). The sample largely comprised young adults with some college education, consistent with the high proportion of students in our recruitment area. Total of 28 participants (26.7%) exceeded the clinical cutoff on the EDE-Q global scale (2.4; [62]) indicating the likely presence of a diagnosable ED. Approximately half of the sample (*n* = 53; 50.5%) endorsed past-month overeating behavior and just under one-third reported past-month LOC eating (*n* = 33; 31.4%) and/or binge eating episodes (*n* = 32; 30.5%). Because of low endorsement of purging behaviors (vomiting: *n* = 7; 6.7% or laxative use: *n* = 2; 1.9%), we summed the frequency of these behaviors with past-month excessive exercise (*n* = 32; 30.5%; *M* = 2.27, *SD* = 4.82) to calculate a composite variable; 36 participants (34.3%) reported engaging in some form of compensatory behavior on this summary measure.

### 3.2. Correlations among Negative Urgency, NEAT, NSSI history, and ED Symptoms

We ran a series of Spearman’s *rho* correlations to determine relationships among negative urgency, NEAT, lifetime NSSI history (absent = 1; present = 2), EDE-Q scales, and EDE-Q behavior items (Table 2). These analyses illustrated that negative urgency was significantly associated with NEAT, NSSI history, and all EDE-Q scale scores. Negative urgency was further associated with behavioral items from the EDE-Q, but not the composite variable of compensatory behaviors. NEAT demonstrated a similar pattern of consistent associations with greater EDE-Q scores across all scales and behavioral items, with the exception of past-month overeating frequency. In contrast to negative urgency, however, NEAT was significantly correlated with more frequent compensatory behaviors. Lifetime history of NSSI was associated with more cognitive ED symptoms (i.e., concerns about eating, shape, and weight) but not with behavioral ED symptoms (i.e., EDE-Q restraint scale scores, excessive eating items, and compensatory behaviors). 

We also examined the associations between demographic variables and constructs of interest, after dummy coding known correlates of EDs and NSSI: biological sex (male = 1; female = 2), gender minority status/sexual orientation (heterosexual = 1; non-heterosexual = 2), and race (white = 1; non-white = 2). We recoded the latter two variables as binary given the relatively low number of participants who identified as LGBTQ+ or as non-white. Analyses revealed that non-heterosexual participants reported higher negative urgency, *rho*(102) = 0.23, *p* = 0.02, and were more likely to endorse NSSI histories, *rho*(103) = 0.36, *p* < 0.001. Participants who identified their race as “white” reported greater weight concerns, and more frequent past-month binge eating episodes, both *rho*(103) = −0.21, *p* = 0.03. We therefore included sexual (gender/orientation) and racial minority status (in addition to biological sex) as binary covariates in the following regression models. We observed no other significant relationships between demographic characteristics and negative urgency, NEAT, or EDE-Q variables.

### 3.3. Hierarchical Linear Regression Analyses Predicting EDE-Q Scales from Negative Urgency and NEAT

We then evaluated a set of hierarchical regressions to assess whether NEAT accounts for overlapping variance in ED symptoms explained by negative urgency. We used linear models to examine relative effects of negative urgency and NEAT on the EDE-Q global scale and all EDE-Q subscales. After controlling for demographic covariates and NSSI history, NEAT additively contributed to negative urgency in predicting EDE-Q global scale scores (Figure 1). Both negative urgency and NEAT were the only significant predictors in the final, three-step model (see Table 3). We observed similar patterns for EDE-Q subscales reflecting cognitive symptoms associated with EDs, with the exception of weight concerns. This discrepancy prompted us to run exploratory alternative models, in which we entered NEAT on Step 2 ahead of negative urgency (entered on Step 3); please see the Appendix A for full results of these analyses. Specifically, the bivariate association between impaired NEAT and EDE-Q weight concerns was no longer evident when controlling for negative urgency (and other factors) in linear regression, confirming that a) these constructs explained overlapping variance in EDE-Q weight concerns and b) negative urgency had additional predictive utility beyond the effect of NEAT. 

Results also differed when we applied a linear model with the same set of predictors to EDE-Q restraint subscale scores. Only NEAT had a main effect in the final model for EDE-Q restraint, which was itself non-significant (see Table 4). This model *was* significant, however, after removing demographic covariates that did not correlate with EDE-Q restraint scores, *F*(3,101) = 7.09, *p* = 0.037, *R*^2^ = 0.08. This more parsimonious model controlled for ESST accuracy (entered on Step 1), *F*(1,100) = 4.95, *p* = 0.95, with negative urgency and NEAT both entered on Step 2, Δ*F*(2,98) = 4.40, *p* = 0.05, Δ*R*^2^ = 0.08. Again, NEAT remained the only significant predictor of EDE-Q restraint in the streamlined model, *B* = 0.02, *SE* = 0.01, β = 0.15, *p* = 0.041 (negative urgency: *B* = 0.04, *SE* = 0.02, β = 0.21, *p* = 0.07).

### 3.4. Zero-Inflated Regression Analyses Predicting ED Behaviors from Negative Urgency and NEAT

Following Schaumberg et al. [69], we used zero-inflated negative binomial regression (ZINB) analyses to examine past-month frequency of disordered eating behaviors, as indexed by items on the EDE-Q (Table 4); likelihood-ratio tests confirmed that ZINB regression provided superior fit to alternatives, e.g., zero-inflated Poisson regression. ZINB analyses of non-zero data revealed that men engaged in more frequent past-month compensatory behaviors (e.g., compulsive exercise or purging) in the final three-step model including all predictors, *B* = 2.74, *SE* = 0.46, incident risk ratio (IRR) = 15.55, *z* = 5.94, *p* < 0.001. Participants who identified as non-heterosexual also reported more frequent compensatory behaviors (*B* = 2.58, *SE* = 0.47, IRR = 13.14, *z* = 5.53, *p* < 0.001) as well as LOC eating episodes, *B* = 1.04, *SE* = 0.48, IRR = 2.83, *Z* = 2.19, *p* = 0.029. Non-white participants similarly endorsed more frequent compensatory behaviors, *B* = 1.11, *SE* = 0.40, IRR = 3.04, *z* = 2.78, *p* < 0.001. No other demographic variables remained significant in full ZINB count models with all predictors. In contrast to cognitive ED symptoms, NSSI history did not significantly increase the likelihood of disordered eating behaviors, but participants *without* NSSI history reported more frequent past-month compensatory behaviors, *B* = −0.94, *SE* = 0.41, IRR = 0.39, *z* = −2.31, *p* = 0.021. We additionally found an effect of ESST accuracy that was unique to overeating frequency (*B* = −0.06, *SE* = 0.02, IRR = 0.95, *z* = −3.40, *p* < 0.001), such that individuals who were less accurate in identifying the emotional valence of images reported more overeating episodes. 

Negative urgency had main effects on all disordered eating behaviors but not compensatory behaviors (see Table 4). We observed additive effects of NEAT beyond negative urgency in two of the four models of behavioral ED symptoms, echoing hierarchical linear regression results obtained for other EDE-Q variables. Specifically, NEAT predicted frequency of overeating (*B* = 0.02, *SE* = 0.01, IRR = 1.02, *z* = 2.18, *p* = 0.03) and loss-of-control eating (*B* = 0.03, *SE* = 0.01, IRR = 1.03, *z* = 2.51, *p* = 0.012) but not binge eating (*B* = 0.02, *SE* = 0.02, IRR = 1.02, *z* = 1.06, *p* = 0.29) nor compensatory behaviors (*B* = −0.03, *SE* = 0.02, IRR = 0.98, *z* = −1.80, *p* = 0.073), after controlling for demographics, NSSI history, and negative urgency. We additionally found an effect of NEAT on EDE-Q binge eating when entered on the second step ahead of negative urgency, *B* = 0.05, *SE* = 0.02, IRR = 1.05, *z* = 2.61, *p* = 0.009 (see Appendix A), comparable to results obtained for EDE-Q weight concerns.

## 4. Discussion

Dysregulated eating behaviors are common among young adults and are associated with pronounced distress and impairment. Advancing our understanding of neurocognitive dysfunction that characterizes different forms of dysregulated eating is critical to refine models of self-destructive behaviors [1]. The present study examined associations among late negative emotional response inhibition (i.e., NEAT), negative urgency, and ED symptoms (cognitive and behavioral). Three primary findings emerged. First, NSSI history was significantly associated with more severe cognitive ED symptoms, as well as *fewer* compensatory behaviors, partially replicating prior research [2,3,4]. Second, worse NEAT (i.e., more negative valence false alarms, reflecting difficulty inhibiting aversive emotional reactions) was associated with more severe cognitive symptoms of ED as well as more frequent overeating and LOC eating, controlling for NSSI and negative urgency (except for weight concerns); however, negative urgency remained a significant predictor of these symptoms, contrary to our hypotheses. Finally, worse NEAT was uniquely associated with the restraint subscale of the EDE-Q. NEAT thus accounted for the effect of negative urgency on restrictive eating but not on other ED behaviors, providing partial support for the hypotheses motivating this study.

NSSI and ED symptoms frequently co-occur, and scholars have proposed that certain ED behaviors may be conceptualized as “indirect” types of self-injury [1,5]. Although often treated as clinically distinct phenomena, NSSI and ED behaviors may share common functions (e.g., causing physical damage to one’s body and providing emotional relief [13,14,15,24]) and risk factors (e.g., impulsive responsivity to negative affect [10,11,12,34,35,36,43,44]). Our findings support the latter possibility, given that both negative urgency and poor negative emotional response inhibition were modestly associated with NSSI and ED symptoms in bivariate analyses. However, in the present study, the link between NSSI and ED symptoms was clearer and more consistent for *cognitive* ED symptoms. We found that lifetime NSSI history was associated with eating, shape, and weight concerns in this non-clinical sample of adults. Although not directly tested here, these associations may reflect lowered body regard (i.e., how one experiences, cares for, and views one’s own body; see [70]) that characterizes both EDs and NSSI [71,72]. Notably, our results are consistent with recent work showing that, among female patients with EDs, those with a history of NSSI reported higher appearance valuation, body dissatisfaction, more negative feelings and attitudes toward their bodies, and less comfort with physical contact than those without NSSI [72]. It is therefore plausible that comorbidities among NSSI and dysregulated eating *behaviors* may be partially explained by the link between NSSI and ED-related *cognitions* (e.g., self-critical evaluations of one’s body shape or weight). Additional research is needed to unpack the degree to which the co-occurrence of NSSI and EDs reflects shared abnormalities in cognition, behavior, or both.

Given the overlap in proposed functions and contributing mechanisms between ED and NSSI behaviors [5,10], we also expected that engaging in one type of self-injury would increase the likelihood of also performing the other. In contrast to this expectation, endorsing a history of NSSI engagement was associated with *less* frequent compensatory behaviors (i.e., compulsive exercise and purging). Prior research suggests that both NSSI and ED behaviors serve to regulate emotion. Negative affect typically precedes NSSI, and the most common outcomes of NSSI include reducing distress and achieving relief [21,22,23,25,30,31,32]. Negative mood is similarly hypothesized to trigger dysregulated eating and compensatory behaviors, and people generally report reduced negative affect following episodes of ED behaviors, e.g., [26,27,28,29]. It is possible that, for some participants in our sample, engagement in behaviors classified as NSSI (i.e., “direct” self-injury) might obviate compensatory behaviors (indirect self-harm) as a strategy for regulating negative affect (and vice-versa), given overlapping motives and outcomes. Recent studies have suggested that pain is more effective at improving mood than cognitive emotion regulation strategies, producing substantial reductions in negative affect, at least in the short-term [20]. Future work should examine the degree to which people with EDs may substitute direct forms of self-harm for compensatory behaviors, in addition to elucidating specific factors that might predispose individuals to choose NSSI or more indirect forms of self-harm when experiencing heightened negative affect. From a clinical perspective, *current* ED features and *lifetime* NSSI history had comparably-sized associations with NEAT. This provides preliminary evidence that impaired negative emotional response inhibition may be a more sensitive marker of emotional difficulties stemming from ongoing eating-related problems compared to active NSSI. Subsequent research ought to determine the extent to which NEAT is amenable to intervention, given its potential utility as a therapeutic target in ED treatment.

Contrary to expectations and our prior findings in NSSI [53], the explanatory contributions of negative urgency and NEAT (to variation in ED symptoms) were largely independent. NEAT deficits added significant variance (3%–7%) to the prediction of global ED symptoms, eating concerns, shape concerns, overeating, and LOC eating, while controlling for negative urgency and NSSI history. This suggests that terminating initiated motor actions triggered by negative emotional reactions may only represent one facet of negative urgency (hence the small correlation). A more comprehensive behavioral assessment of processes involved in *affective control* (i.e., the putative neurocognitive mechanisms underlying emotion dysregulation; see [12]) is needed to identify additional components of urgency and related traits. The finding that impaired emotional response inhibition more strongly relates to cognitive symptoms of EDs may be due to the fact that these aspects of EDs are closely connected to deficits in hot executive functioning and hypoactivity in frontal brain areas implicated in emotion dysregulation, repetitive negative thinking, and self-injurious behaviors [12]. Ultimately, negative emotional response inhibition impairment does not explain the relationship between urgency and dysregulated eating, since each make unique contributions to ED symptoms.

Among ED behaviors, worse NEAT was specifically associated with restrictive eating, while the effect of negative urgency was non-significant. We observed the reverse pattern for binge eating. These results are inconsistent with prior work that generally finds no difference between individuals with *anorexia nervosa* and healthy controls on standard (non-emotional) variants of the stop-signal task, see [73]. This pattern mirrors findings in the NSSI literature, as most studies report no evidence of overall response inhibition impairment among individuals with NSSI histories relative to healthy or clinical control groups [74,75,76,77,78,79]. However, our previous research evaluating *emotional* response inhibition in NSSI suggests a specific deficit in the ability to terminate ongoing motor responses driven by negative affect (i.e., NEAT) that helps account for elevated negative urgency in this population [53]. It remains unclear why difficulty inhibiting emotion-triggered *behavioral* responses is more strongly associated with *cognitive* symptoms of EDs compared to dysregulated eating, as well as with food *restriction* rather than overconsumption, given that “restraint” implies heightened impulse control. Importantly, the EDE-Q restraint subscale assesses *attempts* at food restriction rather than success, suggesting that NEAT dysfunction might possibly enhance the likelihood that individuals will *attempt* to regulate mood by limiting food intake. Difficulty inhibiting negative emotional reactions thus does not necessarily relate to successful inhibition of eating-related impulses (which may or may not be prompted by negative affect). Further empirical work is needed to fully elucidate the role of self-reported impulsive personality traits in cognitive and behavioral symptoms of EDs, as the current study implicates potentially distinct neurocognitive mechanisms for dysregulation in these phenomena. 

Our findings should be considered in light of the following limitations. First, these results must be replicated in a larger, more diverse sample, ideally targeted to include participants endorsing NSSI history without substantial ED symptoms and vice-versa. The present study was sufficiently powered at a level of 0.85 to detect small incremental effects in hierarchical regression; regardless, replicating these effects in participant groups with distinct (e.g., mutually exclusive) histories of NSSI and ED symptomatology would inspire further confidence. Because we recruited a community sample, behavioral symptoms of EDs were mild and most participants would not meet diagnostic criteria for an ED. It remains unclear whether results would be observed among people experiencing clinical levels of these symptoms. Further, purging behaviors (e.g., vomiting) were rarely reported in our sample, while these compensatory behaviors are much more common in diagnosable EDs. It is possible that the relatively low endorsement of ED symptoms, particularly purging, might be an artifact of recruiting community participants without ED treatment histories for a study primarily focused on NSSI. Shame and/or stigma around disclosing such symptoms (in the context of NSSI-related research) may have artificially limited the range of disordered eating severity, consequently hampering our ability to detect relations between negative emotional response inhibition and ED behaviors. Relatedly, the present sample is demographically representative of the catchment area from which participants were recruited. Relative homogeneity among study participants could restrict the generalizability of our results beyond white community adults and college students. These findings therefore encourage subsequent work examining potential mechanisms of negative urgency in ethno-racially diverse clinical populations. Third, we relied on participant self-report to assess ED symptoms and did not measure additional psychiatric features that could partially account for our findings. For instance, alcohol and other substance use disorders often co-occur with EDs, e.g., [6,36] and are associated with deficits in motor response inhibition; see [51,68,80]. Future research that considers the effect of NEAT on ED symptoms in the context of a more comprehensive diagnostic and symptom assessment is warranted, ideally using structured clinical interviews. Finally, given our cross-sectional design, it is unclear whether NEAT deficits are a cause or a consequence of ED symptoms. Some biological consequences of EDs (e.g., malnutrition) can have considerable neurocognitive consequences [81]; thus, longitudinal analyses are needed to confirm that impaired negative emotional response inhibition precedes worsening ED symptoms.

## 5. Conclusions

In sum, the current study did not find evidence supporting our hypothesis that impaired negative emotional response inhibition serves as a mechanism for negative urgency in ED-related cognitions and behaviors. These results contrast with prior work indicating that deficits in this neurocognitive process partially underlie negative urgency in NSSI. However, this research suggests a robust association between NEAT impairment and ED symptoms, independent of negative urgency and NSSI history. Such findings are consistent with substantial evidence indicating that impulsivity and emotion dysregulation may be common factors in the etiology of these frequently comorbid psychopathologies. Poor late-stage negative emotional response inhibition may not represent a common mechanism for negative urgency across direct and indirect self-injurious behaviors, but instead reflect shared neurocognitive dysfunction across NSSI and ED symptomatology. 

## Figures and Tables

**Figure 1 brainsci-10-00104-f001:**
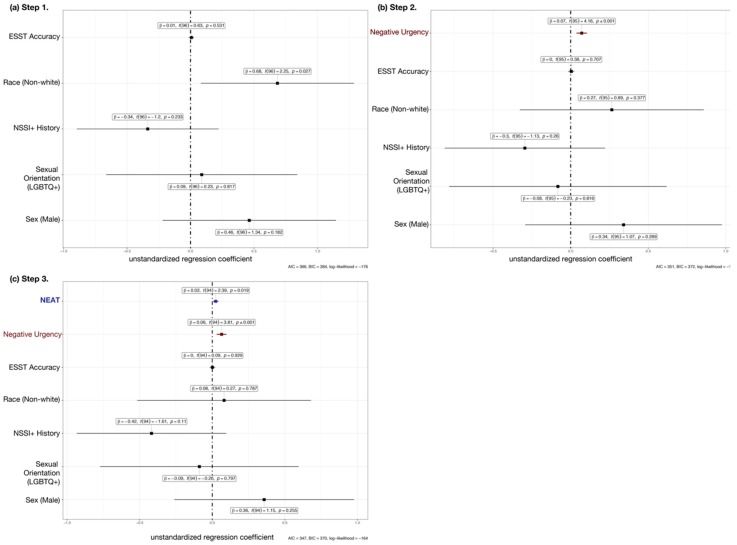
Multiple linear regression results for EDE-Q global scale scores with predictors entered hierarchically in three steps: (**a**) Sex, gender/sexual orientation, race, NSSI history, ESST accuracy (Step 1); (**b**) negative urgency (Step 2); and (**c**) NEAT (Step 3). Both negative urgency and NEAT uniquely contributed significant variance to the prediction of ED symptoms, after controlling for demographic variables, NSSI history, and ESST performance (see Table 3 for additional information). AIC = Akaike’s Information Criterion; BIC = Bayesian Information Criterion.

**Table 1 brainsci-10-00104-t001:** Demographic characteristics (*N* = 105).

		*M (SD)*
	**Age**	24.41 (8.04)
	**Years of Education**	14.42 (1.90)
		***n* (%)**
**Sex**	Female	82 (78.1)
	Male	23 (21.9)
**Gender/Orientation**	Heterosexual	85 (81.0)
	LGBTQ+	20 (19.0)
**Race/Ethnicity**	White	54 (51.4)
	Black	11 (10.5)
	Asian	27 (25.7)
	Hispanic/Latinx	4 (3.8)
	Mixed/Other	9 (8.6)
**Relationship Status**	Single	49 (46.7)
	Dating	39 (37.1)
	Married	10 (9.52)
	Divorced	3 (2.9)
	Cohabitating	3 (2.9)
	Other	1 (0.95)

**Table 2 brainsci-10-00104-t002:** Bivariate non-parametric (Spearman’s) correlations (*N* = 103).

	*M (SD)*	1.	2.	3.	4.	5.	6.	7.	8.	9.	10.	11.
**1. Negative Urgency**	25.52 (8.35)											
**2. NEAT**	40.45 (14.13)	0.26 **										
**3. NSSI History (binary)**	187.42 (519.07) ^a^	0.38 ***	0.27 **									
**4. EDE-Q Global**	1.53 (1.42)	0.35 ***	0.28 **	0.22 *								
**5. EDE-Q Eating Concerns**	0.89 (1.38)	0.45 ***	0.32 ***	0.24 *	0.81 ***							
**6. EDE-Q Shape Concerns**	2.00 (1.68)	0.43 ***	0.27 **	0.28 **	0.93 ***	0.76 ***						
**7. EDE-Q Weight Concerns**	1.73 (1.69)	0.31 ***	0.26 **	0.20 *	0.92 ***	0.68 ***	0.88 ***					
**8. EDE-Q Restraint**	1.52 (1.58)	0.22 *	0.20 *	0.02	0.84 ***	0.66 ***	0.66 ***	0.69 ***				
**9. EDE-Q Overeating**	2.67 (5.18)	0.36 ***	0.16	0.11	0.41 ***	0.44 ***	0.35 ***	0.26 **	0.33 ***			
**10. EDE-Q LOC Eating**	1.87 (4.79)	0.30 **	0.26 **	0.14	0.43 ***	0.57 ***	0.38 ***	0.36 ***	0.30 **	0.60 ***		
**11. EDE-Q Binge Eating**	1.55 (4.28)	0.35 ***	0.36 ***	0.11	0.49 ***	0.58 ***	0.45 ***	0.43 ***	0.40 ***	0.65 ***	0.82 ***	
**12. EDE-Q Comp. Behaviors**	2.78 (6.60)	0.13	0.23 *	−0.02	0.49 ***	0.38 ***	0.44 ***	0.41 ***	0.51 ***	0.17	0.23 *	0.30 **

^a^ Lifetime NSSI episodes among those reporting NSSI history (*n* = 45). NEAT = negative emotional action termination; higher values reflect worse negative emotional response inhibition on the ESST. EDE-Q LOC Eating = past-month episodes of “loss-of-control” eating. EDE-Q Comp. Behaviors = compensatory behaviors composite; sum of past-month vomiting, laxative use, and compulsive exercise episodes. Correlational analyses based on 103 degrees of freedom (*df*) except for those including negative urgency (*df* = 102) because of missing data from one participant. * *p* < 0.05; ** *p* < 0.01; *** *p* < 0.001.

**Table 3 brainsci-10-00104-t003:** Hierarchical linear regression analyses: Cognitive ED symptoms (*N* = 102).

	EDE-Q Global		EDE-Q Eating Concerns		EDE-Q Shape Concerns		EDE-Q Weight Concerns	
**Step 1.**	***B (SE)***	**β**	***B (SE)***	**β**	***B (SE)***	**β**	***B (SE)***	**β**
**Sex ^a^**	0.46 (0.34)	0.13	0.48 (0.33)	0.14	**0.74 (0.40) ***	**0.18**	0.68 (0.40)	0.16
**Gender/Orientation ^a^**	0.09 (0.38)	0.02	0.27 (0.36)	0.08	0.11 (0.44)	0.03	0.09 (0.44)	0.02
**Race ^a^**	−0.34 (0.28)	−0.12	−0.31 (0.27)	−0.11	−0.30 (0.33)	−0.09	−0.58 (0.32)	−0.17
**NSSI History ^a^**	**0.68 (0.30) ***	**0.24**	**0.63 (0.29) ***	**0.23**	**0.95 (0.35) ***	**0.28**	**0.90 (0.40) ***	**0.26**
**ESST Accuracy**	0.01 (0.01)	0.06	0.00 (0.01)	0.03	0.01 (0.01)	0.05	0.02 (0.01)	0.12
**Step 1:**	Δ*F*(5,96) = 2.19, Δ*R*^2^ = 0.10		**2.42 ***, Δ*R*^2^ = 0.11		**2.88 ***, Δ*R*^2^ = 0.13		**3.43 ****, Δ*R*^2^ = 0.15	
**Step 2.**	***B (SE)***	**β**	***B (SE)***	**β**	***B (SE)***	**β**	***B (SE)***	**β**
**Negative Urgency**	**0.07 (0.02) *****	**0.41**	**0.08 (0.02) ***	**0.51**	**0.08 (0.02) *****	**0.40**	**0.07 (0.02) ****	**0.33**
**Step 2:**	Δ*F*(1,95) = **17.33 *****, Δ*R*^2^ = 0.14		**29.74 *****, Δ*R*^2^ = 0.21		**16.93 *****, Δ*R*^2^ = 0.13		**10.98 *****, Δ*R*^2^ = 0.09	
**Step 3.**	***B (SE)***	**β**	***B (SE)***	**β**	***B (SE)***	**β**	***B (SE)***	**β**
**Negative Urgency**	**0.06 (0.02) ****	**0.37**	**0.08 (0.02) *****	**0.46**	**0.08 (0.02) *****	**0.37**	**0.06 (0.02) ****	**0.30**
**NEAT**	**0.02 (0.01) ****	**0.23**	**0.03 (0.01) ****	**0.29**	**0.02 (0.01) ***	**0.19**	0.02 (0.01)	0.17
**Step 3:**	Δ*F*(1,94) = **5.72 ***, Δ*R*^2^ = 0.04		**11.09 *****, Δ*R*^2^ = 0.07		**3.94***, Δ*R*^2^ = 0.03		3.18, Δ*R*^2^ = 0.03	
**Full Model:**	*F*(7,101) = **5.34 *****, *R*^2^ = 0.29		**8.77 *****, *R*^2^ = 0.40		**5.53 *****, *R^2^* = 0.29		**4.82 *****, *R*^2^ = 0.26	

^a^ Binary dummy-coded variables: Sex (Female = 1; Male = 2), Gender/Orientation (Heterosexual = 1; LGBT/Q = 2), Race (White = 1; Non-white = 2), and NSSI History (No = 1; Yes = 2). Parameter estimates and significance values associated with corresponding *t-*tests derived from bootstrapping with 5000 replications; **p* < 0.05; ***p* < 0.01; ****p* < 0.001. Significant effects are highlighted in bold typeface.

**Table 4 brainsci-10-00104-t004:** Hierarchical regression analyses: Behavioral ED symptoms (*N* = 102).

	EDE-Q Restraint		EDE-Q Overeating ^a^		EDE-Q LOC Eating ^a^		EDE-Q Binge Eating ^a^		EDE-Q Comp. Behaviors ^a^	
**Step 1.**	***B (SE)***	**β**	***B (SE)***	**IRR**	***B (SE)***	**IRR**	***B (SE)***	**IRR**	***B (SE)***	**IRR**
**Sex**	−0.05 (0.40)	−0.01	0.53 (0.56)	1.69	**1.44 (0.69) ***	**4.24**	**1.44 (0.58) ***	**4.22**	**2.64 (0.43) *****	**13.96**
**Orient.**	−0.12 (0.44)	−0.03	0.66 (0.59)	1.93	**1.43 (0.69) ***	**4.17**	0.99 (0.54)	2.51	**2.31 (0.41) *****	**10.09**
**Race**	−0.17 (0.33)	−0.05	−0.50 (0.34)	0.61	0.28 (0.49)	1.33	−0.77 (0.51)	0.46	**0.67 (0.32) ***	**1.96**
**NSSI Hx.**	0.27 (0.36)	0.08	0.10 (0.50)	1.10	−0.51 (0.56)	0.60	−0.34 (0.50)	0.71	**−1.14 (0.36) ****	**0.32**
**ESST Acc.**	0.00 (0.01)	0.02	**−0.06 (0.21) ****	**0.95**	0.03 (0.02)	1.03	**0.05 (0.02) ***	**1.06**	0.02 (0.01)	1.02
**Step 1:**	Δ*F*(5,96) = 0.20 Δ*R*^2^ = 0.10		χ^2^(10,89) = 16.47 LL: −195.49		χ^2^(10,89) = **22.74 *** LL: **−139.26**		χ^2^(10,89) = **21.34 *** LL: **−131.74**		χ^2^(10,89) = **54.29 ***** LL: **−153.51**	
**Step 2.**	***B (SE)***	**β**	***B (SE)***	**IRR**	***B (SE)***	**IRR**	***B (SE)***	**IRR**	***B (SE)***	**IRR**
**Negative Urgency**	**0.05 (0.02) ***	**0.27**	**0.05 (0.02) ****	**1.06**	**0.06 (0.03) ***	**1.06**	**0.07 (0.02) ****	**1.07**	0.00 (0.02)	1.00
**Step 2:**	Δ*F*(1,95) = **5.82 *** ΔR^2^ = 0.06		χ^2^(2,87) = **19.44 ***** LL: **−186.08**		χ^2^(2,87) = **12.06 **** LL: **−131.65**		χ^2^(2,87) = **18.93 ***** LL: **−121.60**		χ^2^(2,87) = 1.54 LL: −152.70	
**Step 3.**	***B (SE)***	**β**	***B (SE)***	**IRR**	***B (SE)***	**IRR**	***B (SE)***	**IRR**	***B (SE)***	**IRR**
**Negative Urgency**	0.05 (0.02)	0.23	**0.05 (0.02) ****	**1.05**	**0.06 (0.02) ****	**1.07**	**0.07 (0.02) *****	**1.08**	0.01 (0.02)	1.00
**NEAT**	**0.02 (0.01) ***	**0.19**	**0.02 (0.01) ***	**1.02**	**0.03 (0.01) ***	**1.03**	0.02 (0.02)	1.02	−0.03 (0.02)	0.98
**Step 3:**	Δ*F*(1,94) = 2.94 ΔR^2^ = 0.03		χ^2^(2,85) = 5.28 LL: −183.52		χ^2^(2,85) = **9.86 **** LL: **−129.62**		χ^2^(2,85) = 2.46 LL: −106.11		χ^2^(2,85) = **7.98 *** LL: **−147.33**	
**Full Model:**	*F*(7,101) = 1.42 R^2^ = 0.10		χ^2^(14,85) = **45.78 ***** AIC: 401.04		χ^2^(14,85) = **47.51 ***** AIC: 293.25		χ^2^(14,85) = **41.93 ***** AIC: 246.22		χ^2^(14,85) = **57.65 ***** AIC: 328.65	

Orient. = gender/sexual orientation (Heterosexual = 1; LGBT/Q = 2); NSSI Hx. = NSSI history (*No* = 1; *Yes* = 2); ESST Acc. = percent accuracy of stimulus categorization (by valence); IRR = incident risk ratio; LL = log-likelihood. ^a^ Zero-inflated negative binomial regression results for count models (see Appendix A for logistic zero-inflated model results). Chi-square values obtained at each step via Wald tests; full model chi-square derived from comparison against null (constant-only) model. * *p < 0*.05; ** *p < 0*.01; *** *p < 0*.001. Significant effects are highlighted in bold typeface.

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
