# Peer review of "Emotional Response Inhibition: A Shared Neurocognitive Deficit in Eating Disorder Symptoms and Nonsuicidal Self-Injury"

_brainsci, 2020, doi:10.3390/brainsci10020104_

Round 1
Reviewer 1 Report
Authors present an interesting and innovative study of clinical significance, in which they examine potential shared mechanisms- specifically emotion response inhibition, to eating disorder symptoms and nonsuicidal self-injury (NSSI). Strengths of the paper include addressing a question of clinical significance, examining the question of a potential shared biomechanism involved in both eating disorder and NSSI, sound methodological design and data analytic strategy, and authors present the rationale, significance, and theoretical basis for the study clearly in the introduction.
Potential points to address:
How do authors envision shame/stigma around self-disclosing, sharing eating disorder symptoms and purging behaviors as impacting study recruitment and the sample studied? From clinical experience and research knowledge, I imagine that some individuals with severe ED symptoms who have also not previously presented for intervention in the community, may be secretive and private about their purging behavior. This seems consistent with a limitation the authors identify in the discussion- that purging/vomiting behaviors were lowly reported, in comparison to compensatory behaviors. Authors may benefit from more specifically discussing how low level of purging/ vomiting may potentially be an artifact of recruitment. Authors would benefit from addressing the sample demographics as a potential limitation and need for examining these questions and potential shared mechanisms in ethnoracially diverse and clinically high-risk samples.Author Response
Please see the attachment.

Reviewer 2 Report
This study is both generative and creative. The manuscript is written well and was a pleasure to read. The rationale for the study is clear, the method and results are appropriate, and the interpretation is compelling. Although the authors did not find support for their primary hypothesis, the association of NEAT dysfunction with ED is important.
There are two additions that would make this manuscript stronger.
First, I would like to see the authors address the issue of power. Given that the authors did not find support for their primary hypothesis, I would like to know whether they had the power to detect a significant effect. In other words, did they fail to find an effect because one does not exist, or because of sample size limitations? This is easily addressed, and its addition would bolster the strength of the manuscript. However, I would be pleased to see this published even without that analysis. Second, it would be useful for the authors to interpret their findings in light of clinical implications. I realize that the ED behaviors were relatively minor in this sample. Still, it seems that this study might have insights for clinical practice and treatment research. This is less important in my mind than adding a power analysis, particularly given the typically non-applied audience that Brain Sciences attracts.Author Response
Please see the attachment.
